# The Effect of Diabetes Mellitus on IGF Axis and Stem Cell Mediated Regeneration of the Periodontium

**DOI:** 10.3390/bioengineering8120202

**Published:** 2021-12-03

**Authors:** Nancy M. S. Hussein, Josie L. Meade, Hemant Pandit, Elena Jones, Reem El-Gendy

**Affiliations:** 1Division of Oral Biology, School of Dentistry, University of Leeds, Leeds LS2 9JT, UK; dnnmsh@leeds.ac.uk (N.M.S.H.); j.l.meade@leeds.ac.uk (J.L.M.); 2Department of Oral Medicine and Periodontology, Faculty of Dentistry, Mansoura University, Mansoura 35516, Egypt; 3Leeds Institute of Rheumatic and Musculoskeletal Medicine, School of Medicine, University of Leeds, Leeds LS2 9JT, UK; H.Pandit@leeds.ac.uk (H.P.); e.jones@leeds.ac.uk (E.J.); 4Department of Oral Pathology, Faculty of Dentistry, Suez Canal University, Ismailia 41611, Egypt

**Keywords:** diabetes, periodontal disease, mesenchymal stem cells, bone regeneration

## Abstract

Periodontitis and diabetes mellitus (DM) are two of the most common and challenging health problems worldwide and they affect each other mutually and adversely. Current periodontal therapies have unpredictable outcome in diabetic patients. Periodontal tissue engineering is a challenging but promising approach that aims at restoring periodontal tissues using one or all of the following: stem cells, signalling molecules and scaffolds. Mesenchymal stem cells (MSCs) and insulin-like growth factor (IGF) represent ideal examples of stem cells and signalling molecules. This review outlines the most recent updates in characterizing MSCs isolated from diabetics to fully understand why diabetics are more prone to periodontitis that theoretically reflect the impaired regenerative capabilities of their native stem cells. This characterisation is of utmost importance to enhance autologous stem cells based tissue regeneration in diabetic patients using both MSCs and members of IGF axis.

## 1. Introduction

Periodontitis is one of the most common chronic inflammatory diseases that can lead to destruction of the tooth attachment apparatus (periodontium), possibly leading to loss of teeth [1]. Type 2 Diabetes Mellitus (T2DM) is one of the most prevalent metabolic disorders characterized by chronic hyperglycaemia [2]. Chronic hyperglycemia leads to accumulation of advanced glycation end products (AGEs), oxidative stress and promotion of systemic inflammation. This might explain the growing evidence about the “two way relationship” between T2DM and Periodontitis [3]. Currently, periodontal therapies including guided tissue regeneration are used to treat severe cases of periodontitis involving alveolar bone defects. These treatments are successful but can be associated with complications that lead to treatment failure such as infections and loss of membrane coverage with higher failure rates in diabetics [4]. The evolving field of stem cell-based regenerative therapies with stem cells and signaling molecules holds great promise for future therapies. However, these approaches require a much better understanding of how these elements will behave under disease conditions such as diabetes and periodontitis.

IGF1 bears 50% homology to insulin and have the same hypoglycaemic effect, with many studies linking fluctuation in IGF1 levels with metabolic disorder and insulin resistance [5]. Insulin Growth Factor (IGF) axis plays a crucial role in osteogenic differentiation [6]. In previous studies from our group we have proven a role of IGFBP-2, BP-3 and BP-4 in osteogenic differentiation of stem cells isolated form dental pulp (DPSCs) [7,8,9]. Hence it seemed crucial to review how mesenchymal stromal cells (MSCs) and members of the insulin-like growth factor (IGF) axis performance under T2DM condition and whether these changes can be reversed for cell- based regenerative therapy as a mode of treatment for periodontitis.

## 2. Periodontitis

Periodontitis is a chronic inflammatory disease that can lead to different degrees of destruction of periodontal tissues (the alveolar bone, the cementum covering root surfaces, the periodontal ligament (PDL) in-between bone and cementum and the gingiva) [1]. It is caused by accumulation of dental plaque containing pathogenic bacteria in close proximity to periodontal tissues. Bacteria and their products activate a host inflammatory response and both lead to an extensive production of proinflammatory cytokines that ultimately causes damage to the collagen fibres and alveolar bone through the recruitment of immune cells and uncoupling of bone remodelling [10,11]. Periodontitis is diagnosed through measuring the tooth clinical attachment loss (CAL), radiographic alveolar bone loss (ABL) and periodontal pocketing [12]. Sever periodontitis is estimated to affect around 10% of adults world population on average, a proportion that did not show significant change between 1990 and 2010 [13]. Polymicrobial synergy and dysbiosis (PSD) model was proposed as an alternative to the classical classification of pathogenic bacterial species in subgingival plaque into red, orange, yellow and green complexes [14] based on cluster analysis, community ordination and correlation to parameters reflecting severity of periodontal disease such as probing depth (PD) and bleeding on probing (BOP). For instance, members of red and orange complexes are usually identified together with higher colonization of orange complex bacteria associated with and preceding colonization of higher numbers of red complex bacteria. It is the red complex that was closely associated with markers of advanced periodontal destruction mentioned above. This model, however, was later challenged by several findings. For instance, periodontitis associated bacterial populations were shown to be more diverse, and species like *Filifactor alocis* was associated with periodontitis at least equally to red complex species [15]. PSD model suggests that different members of oral microbiota play different and specific roles to transform it into a disease causing microbiota. This dysbiotic community evoke, tolerate and benefit from host inflammatory response initiating a vicious circle for periodontal disease continuation [16].

The host cellular inflammatory response in periodontitis goes through different phases including neutrophil followed by macrophages and dendritic cells recruitment leading to the release of several proinflammatory cytokines such as interleukin (IL) -1β, IL-17 and tumour necrosis factor α (TNF-α), matrix metalloproteinase (MMPs) and activation of receptor activator of nuclear factor kappa-Β ligand (RANKL) pathway [17]. Antibodies produced by B-lineage cells diffusing into periodontal pocket or remaining within the periodontal tissues can in theory exert some protective effects. However, antibody dependant activation of immune cells leads to further inflammation and tissue damage. The activated B-lineage cells can also express MMMs, RANKL, IL-1, IL-6 and TNF contributing to soft and hard tissue damage as well [18].

Unlike other types of oral bone defects such as extraction sockets and periapical lesions that undergo healing through self-regeneration under proper clinical care [19], bone and subsequent PDL loss in periodontal disease is irreversible [20].

The evidence of association between periodontal disease and systemic diseases has been growing steadily and exponentially since late 1980s through cross-sectional, longitudinal and interventional studies, and ultimately the term periodontal medicine was coined describing how periodontal disease influence extra oral health [21]. In fact, periodontitis has been suggested to be linked with 57 health problems [22], with the bidirectional relationship of periodontitis and diabetes representing the hot topic of this field.

## 3. Diabetes

Diabetes mellitus (DM) represents a heterogeneous group of chronic metabolic diseases manifested clinically by hyperglycaemia due to lack of insulin secretion, function or both. It is estimated that globally 422 million people had DM in 2014 with 20–69% expected increase by 2030 [23,24]. The major types of DM are type 1 DM (T1DM) and type 2 DM (T2DM) [25].

Low grade chronic inflammation is associated with development and progression of T2DM. Expanding adipose tissue in obesity—a key risk factor of T2DM-produces a number of inflammatory cytokines such as IL-1, 6, 10, TNF-α, angiotensinogen and adiponectin, with some of these cytokines predictive of developing T2DM [26,27]. TNF-α can cause insulin resistance in different ways, such as reducing glucose transporter 4 in adipocytes and insulin receptor signalling pathways. TNF-α also promotes production of C reactive protein (CRP), another inflammatory marker, by hepatocytes and adipocytes [28].

Advanced glycated endproducts (AGEs) represent a varied group of complex molecules found irreversibly in serum and tissues [29]. Glucose interacts with proteins forming AGEs that accumulate and bind to their receptors on osteoblasts causing increased oxidative stress, RANKL expression and osteoclasts activation. Moreover, osteoblasts apoptosis is induced with the overall result of decreased bone mineralization and repair [30]. AGEs also induces more damage of β cells and is linked with insulin resistance [2].

The hyperglycaemia increases production of superoxides and other reactive oxygen species (ROS) in mitochondria. This oxidative stress leads to cell and tissues destruction and contributes to inflammation. ROS also increases AGEs formation, receptors for AGEs (RAGEs) expression and NF-ĸB pathway activation. This is associated with impaired insulin function and late diabetic complications [31].

The number of circulating endothelial progenitor cells (EPCs) that promote angiogenesis and vascular healing was reported to be reduced in DM patients. This could be due to lowered release from bone marrow, shorter survival in circulation, homing outside circulatory system or a combination of all. DM also shifts EPCs differentiation potentials to an inflammatory phenotype [32], which has been characterized as EPCs enhanced ability to endocytose, stimulate naïve T cells and produce IL12 [33].

## 4. Reciprocal Interaction between Diabetes and Periodontitis

As mentioned earlier, DM and periodontitis share a two-way relationship where they affect each other reciprocally and adversely [34]. DM is an established risk factor for periodontitis and the morbidity and severity of periodontitis positively correlates with poorly controlled or long standing diabetes [35], and such pathological influence is even detectable before clinical diagnosis of DM, where prediabetes is associated with higher incidence, prevalence and severity of periodontitis [36].

Diabetes contributes to periodontitis through multiple mechanisms, one of which is altering the oral microbiota to a more pathogenic composition with increased levels of bacterial species associated with periodontitis or poor periodontal healing in diabetic animals, and this ‘diabetic’ microbiota enhanced IL-6 production, osteoclasts differentiation, periodontal bone loss and inflammation when transferred to normoglycemic mice [37], and such microbiological changes can be seen even in both clinically healthy and resolved periodontal sites in T2DM patients [38]. Diabetes is also suggested to alter the host response to the oral bacterial biofilm in multiple ways. Elevated levels of proinflammatory cytokines IL-1β, IL-6, TNF-α and RANKL have been detected within the gingival tissues of diabetics [39]. Another way is stimulation of immune cells with AGEs leading to higher levels of proinflammatory cytokines secretions with periodontal tissues expressing RAGEs as well. Increased collagenase activity and reduced collagen synthesis are possible contributors [40]. High glucose (HG) was shown to induce Toll like receptor 2 (TLR-2), NF-ĸB pathway and IL-1β in human gingival fibroblasts [41] and RAGEs were upregulated on mRNA level in inflamed gingival tissues of T2DM periodontitis patients compared to their non diabetic periodontitis patients [42]. Conversely, DM is linked with reduced production of anti-inflammatory cytokines such as IL-4, IL-10 and transforming growth factor β (TGF-β). The elevated levels of proinflammatory cytokines cause recruitment and activation of neutrophils with further tissue damage. Moreover, DM increases apoptosis of osteoblasts and PDL cells and is linked with upregulation of apoptosis regulating genes [43]. All of the above mentioned changes cause exacerbated clinical presentation of periodontitis even with a subgingival microbiome that has not changed much from healthy [38]. 

The effect of periodontitis on glycaemic control is evident in non-diabetic subjects with periodontitis as they show higher levels of hemoglobin A1c (HbA1C) and fasting blood glucose (FBG) levels and are at higher risk of developing T2DM compared to those with better periodontal status [44], and their HbA1C values positively correlated with CRP serum levels, PD and BOP [45]. The serum levels of proinflammatory cytokines compared to anti-inflammatory ones was high in patients with periodontitis and even higher in patients with periodontitis and T2DM [46]. This is associated with insulin dysfunction, insulin resistance and ensuing hyperglycaemia [47]. Periodontitis has been linked to higher risk of CVD as well [48]. In addition to elevated systemic inflammatory markers, the ‘leakage’ of periodontal pathogens through gingival ulceration into bloodstream stimulates atherogenesis. This could happen through bacteremia or by using blood cells as vehicles [11]. Indeed, P. gingivalis was found to be the most abundant bacteria in non-atherosclerotic human coronary vessels [49] and it contributed to atheroplaque formation independently of dietary lipids in mice [50]. 

On the other hand, proper periodontal therapy has been shown to lower HbA1c levels in T2DM patients by 3–4 mmol/mol (0.3–0.4%), which is equivalent of adding a second antidiabetic drug, but without additional renal and hepatic pharmacological load [51]. Periodontal therapy also reduced serum levels of CRP, IL-1β, IL-6 and TNF-α in diabetic T2DM patients [52,53] and their risk of developing cardiovascular disease (CVD) [54].

## 5. Current Periodontal Therapies in Diabetics

### 5.1. Non-Surgical Periodontal Therapy

Non-surgical or conventional periodontal treatment uses mechanical instrumentation (scaling and root planning, SRP) for removal of calculus and subgingival plaque. Systemic antibiotics, locally used anti-plaque agents such as chlorhexidine, oral hygiene instructions and patient motivation are valuable adjunctives [55]. Non-surgical periodontal therapy aims at arresting periodontal inflammation and disease progression by removal of causative microorganisms and inflamed/necrotic tissues [56]. This therapy is considered the cornerstone and a preliminary part of any periodontal therapy [57] and there is some evidence that it may alleviate the need for periodontal surgery in lesions with PD up to 6 mms [58]. Non-surgical periodontal therapy is especially valuable in T2DM patients where it is linked with considerable reduction in HbA1C and inflammatory cytokines levels as mentioned earlier [59]. Moreover, surgical intervention is not always possible in diabetic patients and several factors have to be considered such as level of glycaemic control, association of CVD, delayed wound healing—a problematic feature in diabetics [2]-in addition to the physical and emotional stresses associated with surgery [60].

Reports of complications of nonsurgical periodontal therapy in diabetics are mixed. T2DM patients with HbA1C levels ranging from 6.5% to 11% and without major diabetic complications treated with non-surgical periodontal therapy and systemic antibiotics showed less gain of clinical attachment, higher risk of gingival recession and higher proportions of periodontal pathogens compared to non-diabetics [61]. In a similar setting (SRP but without antibiotics), no difference in clinical, immunological and microbiological outcomes was detected in diabetics versus non-diabetics. However, T2DM patients included in the study had slightly better glycaemic control (HbA1c between 4.4% and 10.6%) compared to the study mentioned above, and it is possible this could have contributed to these fairly positive outcomes [62]. A systematic review of similar studies concluded that DM does not seem to influence the PD reduction or CAL gain following non-surgical periodontal therapy. However, the follow up period of included studies was up to 6 months only and the outcomes did not include assessment of bone regeneration which is not very commonly seen following non-surgical therapy [63].

Conservative/non-surgical periodontal treatment is known to reduce the microbial load, which can improve the periodontal condition and has a positive effect on controlling blood sugar levels. It was also found that non-surgical treatments can partially improve the gut microbiome and intestinal barrier in experimental animals with hyperlipidemia (64). However, neither surgical nor oral microbiome gut/periodontal therapy is not able to fully alleviate diabetes associated pathology. DM patients are still at higher risk of infection and this line of therapy is not suitable for treating advanced periodontal case with massive tissue loss and damage [64].

### 5.2. Surgical Periodontal Therapy

Surgical periodontal therapy included both ‘historical’ non-regenerative approaches that are hardly used in the present times such as pocket elimination with osseous resection and modified Widman flap with pocket closure [58] as well as the surgical regenerative techniques that aim at restoring the form and function of periodontal tissues. These include bone grafts, root surface conditioning, the use of enamel matrix derivatives (EMD) and guided tissue regeneration (GTR) by applying barrier membranes beneath the soft tissue flap to prevent epithelial cells from contacting root surface and allow for periodontal progenitor cells to fill the wound and regenerate periodontal tissues. The outcome of these regenerative procedures is evaluated clinically (measuring PD and CAL which are indicative of soft tissue healing) and radiographically to assess bone healing with surgical re-entry rarely used [65].

The majority of periodontal regenerative surgeries have been almost exclusively used in infrabony pockets with angular bone loss, where the anatomy of a well contained bone defect particularly increases the success rate of these procedures. A meta-analysis excluding studies on furcation involvement and non self-contained infrabony defects concluded that GTR and EMD achieved better improvements in PD and CAL compared to mere open flap debridement (OFD) surgery in long term follow-ups (up to 5–10 years) [66]. Still, infrabony defects are a fraction of encountered periodontal lesions compared to suprabony defects which pose a greater challenge clinically with rather unpredictable treatment outcomes due to the horizontal pattern of bone loss [67]. In fact, a review published in 2010 reported that horizontal bone loss involved around 92.2% of radiographically examined periodontal lesions, yet paradoxically, only 3.7% of published papers on regenerative periodontal therapy addressed this overwhelming majority [68].

Many factors are in play for the paucity of regenerative periodontal surgeries in suprabony defects. The horizontal bone loss associated with suprabony defects means there are hardly any remaining bony walls to support the mucoperiosteal flap or to provide the vascular and cellular resources needed for healing [69]. Moreover, most membranes used currently in GTR do not have the required mechanical properties to withstand forces transmitted through the flap with possible membrane collapse, loss of healing space and minimal to no osseous regeneration [70]. To overcome these limitations, regeneration in suprabony defects in most instances relied on OFD in combination with growth factors, most notably EMD which resulted in further clinical and radiographic improvement compared to OFD alone [71]. This means that while the benefits of regenerative periodontal surgeries seem evident in infrabony pockets, more research is needed to make these techniques applicable in suprabony pockets. Furthermore, DM patients are at higher risk of post-surgical infection and impaired wound healing [72].

### 5.3. Regenerative Surgical Periodontal Therapy in Diabetic Animal Models

Studies discussed in this section used single intraperitoneal injection of streptozotocin (STZ) to induce diabetes followed by measuring of blood glucose levels—usually 1 week later—to confirm diabetes onset. Additionally, animals with controlled diabetes were also used in some studies and those received a subcutaneous sustained-release insulin implant which was aseptically placed in the dorsal side of the animals’ neck. All studies used male Wistar rats except the work by Lee et al. [73] where female Sprague-Dawley rats were used.

GTR in rats formed new bone at similar rates in healthy, controlled diabetic and uncontrolled diabetic animals, with the later showing higher rate of infections and outcome variation [74]. Diabetic rats treated by applying EMD (EMDOGAIN^®^) into surgically created bone defects showed less bone fill and density and more osteoclasts. EMD enhanced only bone fill in diabetic rats but stimulated bone fill, density and new cementum formation in non-diabetic rats [75]. In a different study where EMD application was preceded by root surface planning and conditioning, diabetic rats had higher rates of bacterial invasion, bone fracture, inflammatory infiltrate, apical migration of attachment epithelium and gingival recession. It is worth mention that in both diabetic and non-diabetic rats, new bone but not new cementum was detectable and EMD had no effect on either [76]. On the contrary, EMD showed better bone healing compared to control sites in diabetic and non diabetic rats. Still diabetic rats had slower bone regeneration and sparse fibres between root surface and new bone compared to non diabetic [77].

Bone graft and GTR in diabetic animal models were tested in skeletal and skull bones with varied results. In one study, commercial porcine cortical-lamellar bone graft, collagen gel and a collagen membrane were applied in tibial defects in diabetic and non diabetic rats. Histologic examination of graft area revealed more graft resorption and new bone formation in healthy animals but the significance of these results was not clearly evaluated [78]. When GTR was tested in calvarial bones of rats using titanium domes, there was no difference in new bone formation between healthy, controlled and uncontrolled diabetic animals [79]. In the same model, osseous healing following titanium disc placement was associated with higher levels of proinflammatory cytokines in diabetics, although this did not equate with difference in bone formation [73]. These findings could indicate the response of diabetics to regenerative periodontal surgeries could be unpredictable and in need of enhancement, especially with the very few clinical studies as shown in the next section.

### 5.4. Regenerative Surgical Periodontal Therapy in Diabetic Patients

Clinical trials of regenerative periodontal surgeries in T2DM patients are not very abundant. One study reported EMD being successfully used in combination with autogenous bone graft in a case-report of a 66 years old T2DM patient [80]. Minimally invasive periodontal surgery with or without EMD was tested in T2DM and non diabetic elderly patients with infrabony pockets. There was no significant difference in bone fill or attachment gain between both groups after 3 years of follow-up [81]. Only recently a study used split mouth technique in controlled T2DM patients with bilateral infrabony pockets to test flap surgery vs surgery with EMD application. Both sides showed clinical improvement after 6 months of follow up with EMD side showing enhanced reduction in PD and gain in the PDL attachment. However, the study did not include non diabetic controls [82].

The scarcity of reports on surgical regenerative periodontal therapy in diabetic patients despite the guidelines indicating that these patients can receive this kind of treatment safely and comparably to non diabetics [60], raises questions about the feasibility of this approach in the real clinical settings. Taken all together, periodontal tissue engineering can provide a more promising alternative therapy for diabetics.

## 6. Tissue Engineering and Periodontal Regeneration

The concept of tissue engineering was first introduced in 1993 and involves the interaction of 3 main elements: cells, signalling molecules including growth factors and matrices or scaffolds for tissue regeneration [83]. This approach is both promising and challenging in regenerating periodontal tissues with their complex soft and hard tissue architecture, the replication of specific angulation of PDL fibres and the micron-scaled vicinity around teeth [84], and DM can further complicate this task. The next section will discuss MSCs and IGF axis as ideal candidates for cells and signalling molecules, respectively.

### 6.1. Stem Cells in Regenerative Periodontal Therapy

Stem cells are defined as cells capable of self-renewal and differentiating into multiple different cell types [85]. There is a plethora of stem cells that are currently being investigated for the purpose of tissue regeneration and cell therapies. They can be broadly classified into: Embryonic stem cells (ESCs), induced pluripotent stem cells (iPSCs) and adult or postnatal stem cells. The use of ESCs and iPSCs remain controversial due to ethical considerations, demanding culturing techniques and risk of tumorigenicity [86], which explains the relative paucity of studies using human ESCs.

To decrease risk of genetic instability and tumour formation of iPSCs, transfection using non-viral vectors (plasmid, proteins and mRNA among others) [87] and inducing their differentiation into one of their downstream lineages such as MSCs before clinical use were proposed [88]. iPSCs have been derived from different types of human dental MSCs including PDL cells which showed superior osteogenic capacities in animal models compared to iPSCs from gingival cells [87]. Mouse iPSCs loaded on silk scaffolds with EMD have successfully induced regeneration of periodontal tissues (more alveolar bone and cementum with PDL fibres in-between compared to scaffolds + EMD controls) in animal models [89]. Moreover, mouse ESCs and iPSCs were shown to differentiate into osteoblasts with similar upregulation of osteogenic markers [90].

Adult or postnatal stem cells are the ones found in postnatal tissues and they also show self-renewal and differentiation capabilities albeit less than ESCs. According to International Society for Cell & Gene Therapy (ISCT), for stem cells to be considered MSCs they should fulfil the following criteria: firstly they must adhere to plastic under standard culture conditions, second they must positively express CD73, CD90 and CD105 and negatively express CD14 or CD11b, CD19 or CD79α, CD34, CD45 and Human Leukocyte Antigen–DR isotype (HLA-DR) and thirdly MSCs must have the capability of multilineage differentiation (osteogenic, chondrogenic and adipogenic) [91]. MSCs have been isolated from various tissues such as bone marrow (bone marrow mesenchymal stromal cells, BM-MSCs), adipose tissue and different dental tissues. Dental stem cells were isolated from several tissue including: dental pulp stem cells (DPSCs), periodontal ligament mesenchymal stem cells (PDL-MSCs), stem cells of human exfoliated deciduous teeth (SHED), stem cells of apical papilla (SCAP) and dental follicle stem cells (DFSCs) [92]. Other types include alveolar BM-MSCs, gingival mesenchymal stromal cells (G-MSCs) and tooth germ derived progenitor cells (TGPCs) [93]. Dental MSCs gained a well-earned interest because they are relatively accessible. Freshly extracted teeth—that are normally discarded in dental clinics—are a source of a different types of dental MSCs [94]. In particular, sound third molars and premolars extracted due to impaction or orthodontic reasons, respectively, are valuable sources of ‘healthy’ stem cells [95]. Most of these cells have been investigated as possible candidates for periodontal regeneration.

Out of these different MSCs, BM-MSCs and PDL-MSCs represent the best candidates for stem cell based periodontal regeneration for different reasons. PDL-MSCs are the native MSCs of PDL tissues and can differentiate into osteoblasts, fibroblasts and cementoblasts with subsequent regeneration of periodontal tissue complex [56]. Periodontal tissue regeneration achieved through stimulating endogenous stem cells using scaffolds, growth factors and drugs would overcome the costs and risks associated with stem cells isolation, expansion and transplantation [96]. BM-MSCs, on the other hand, are considered the gold standard for cellular regenerative therapy and can be harvested from multiple donor sites in relatively large numbers. This means minimum need for expansion and versatility of applications including banking for future use [97] and both cell types have shown immune modulatory characteristics [98,99,100]. Therefore, both will be covered in the next sections.

#### 6.1.1. Periodontal Ligament Mesenchymal Stromal Cells (PDL-MSCs)

PDL-MSCs were first isolated in 2004 through their clonogenic potentials and positive expression of 2 early MSCs markers STRO-1 and CD146/MUC18. The expanded PDL-MSCs displayed in vitro multilineage differentiation into osteoblasts and adipocytes and when transplanted into animal models, they produced PDL like tissue with dense collagen I bundles that were nested into newly formed cementum resembling Sharpey’s fibres [101]. In addition, PDL-MSCs were shown to differentiate into chondrocytes, cardiac myocytes, Schwann cells, astrocytes and retinal ganglion cells and pancreatic cells [102]. PDL-MSCs sustain the morphology (fibroblast like cells with oval nuclei) [103] and the surface markers expression profile of MSCs mentioned earlier [104]. They also contain colony forming cell population [105] and express Nanog and Oct-4, which are embryonic stem cell markers that control self-renewal and pluripotency [106]. In addition to multilineage differentiation, PDL-MSCs undergo self-renewal more than 100 population doublings, with mechanical loading as a possible contributor to this high proliferative capacity [107]. 

Under osteogenic culture conditions, PDL-MSCs form calcified nodules, upregulate early osteogenic markers–markers responsible for osteoblasts differentiation (Runx2, and OSX), bone matrix proteins (Osteocalcin, OCN and Osteopontin, OPN), alkaline phosphatase (ALP) and bone sialoprotein (BSP) [105,108,109]. The calcium deposits of PDL-MSCs have nodular pattern with high density central areas and relatively lower mineral-to-matrix ratio [110]. Osteogenic induction also changes PDL-MSCs morphology into polygonal cells with extended cytoplasmic processes as intercellular bridges which is consistent with osteogenic lineages [111]. The nature of PDL tissue as a highly organized collagenous tissue with well oriented and dense fibres bundles acting as a shock absorbent of physiologic mechanical stresses is reflected in PDL-MSCs expression profile. PDL-MSCs express Scleraxis, a tendon cells specific transcription factor, more than BM-MSCs or DP-MSCs [101]. PDL-MSCs also express a specific isoform of Periostin, a major extracellular matrix (ECM) protein involved in periodontal homeostasis, that was further upregulated under osteogenic conditions [112]. A subset of PDL cells express Cemp-1, a cementoblasts marker, coinciding with increased ALP activity [113]. PDL-MSCs under osteogenic induction displayed upregulation of both osteogenic and cementoblastic markers [114] and also upregulation of osteogenic markers and down regulation of cementoblastic ones. PDL cells overexpressing Cemp-1 exhibited lower mRNA levels of Runx2, OCN and Periostin [113].

PDL-MSCs have immunomodulatory effects as well. PDL-MSCs inhibited peripheral blood mononuclear cells (PBMCs) proliferation through cell cycle inhibition rather than cell death, with IFN-γ produced by PBMCs as the main mediator of PDL-MSCs immunosuppressive influence [115]. PDL-MSCs also supressed the proliferation of T cells through decreasing the expression of CD1b on dendritic cells [116]. This suppressive effect on T cell was sustained after osteogenic differentiation [117]. Aging and passaging could influence PDL-MSCs features. Donor aging was found to suppress most of PDL-MSCs properties, including proliferation, migration, expression of stemness markers STRO-1 and CD146 as well as osteogenic potentials (ALP activity and expression of OCN, OPN and BSP) [105]. Donor aging also increased expression of proinflammatory cytokines IL-1β and IL6 and suppressed anti-inflammatory cytokine IL-4 [118]. Passaging decreased ALP activity and percentage of STRO-1 positive cells but not the expression of osteogenic markers [119].

PDL-MSCs are thought to be available only through dental extraction [104], but SRP of periodontally affected teeth has been proposed as a possible source [103]. However, PDL-MSCs in this instance would be ‘inflamed’ rather than healthy cells. Inflamed PDL-MSCs have the advantage of being readily available without extraction, however they display higher proliferative and lower osteogenic differentiation potentials compared to their healthy counterparts [120]. Attempts to overcome these limitations include adding patient matched healthy PDL-MSCs conditioned medium [121] or co-culturing of patient matched inflamed and healthy PDL-MSCs [122]. However, these cells may still carry the risk of infection and the full effect of periodontal pathogens on PDL-MSCs is not fully clear [104]. For instance, P. gingivalis lipopolysaccharide (LPS) decreased ALP activity, OCN and Col-1 expression in PDL-MSCs [123]. LPS also reduced PDL-MSCs proliferation and expression of vascular endothelial growth factor (VEGF) [124] and stimulated expression of TLR4 and NF-ĸβ [125].

As part of PDL-MSCs identification and isolation, they were transplanted into animal models with periodontal bone defects and were shown to induce periodontal regeneration as mentioned earlier [101,126]. Such technique was moved on into the clinical settings as well. Autologous PDL-MSCs cell sheets with β-TCP improved clinical and radiographic periodontal parameters in 10 patients with periodontitis without any adverse events identified [127]. In a randomized clinical trial using GTR and Bio-oss^®^ bone graft with and without autologous PDL-MSCs cell sheets for treatment of infrabony defects, bone height was increased in both case and control groups without any safety issues regarding PDL-MSCs transplantation. There was, however, no significant difference when PDL-MSCs were used [128]. In a quasi-randomized controlled phase II clinical trial, autologous PDL-MSCs added to xenogeneic bone grafts were safe but did not show additional value compared to bone grafts alone [129]. These results may suggest that stimulation of endogenous PDL-MSCs could be a more practical approach, and that autologous PDL-MSCs may need similar manipulation as well to enhance success rates. 

#### 6.1.2. Bone Marrow Mesenchymal Stromal Cells (BM-MSCs)

BM-MSCs were first observed in guinea pigs bone marrow monolayer cultures in 1970 [130], and were later proven to have the criteria required for MSCs definition proposed by the ISCT [131] which were mentioned earlier. Indeed, positive expression of surface markers CD73, CD90, CD105 and CD40 generally define BM-MSCs with variable colony forming and multilineage differentiation potentials, with surface markers CD271 and CD146 positively selecting cells known to possess higher levels of these potentials [132]. Multiple clinical trials using BM-MSCs for bone regeneration with and without scaffolds have been carried out [133]. In a feasibility and safety trial, autologous iliac crest BM-MSCs were expanded and mixed with bioceramics before being surgically applied to non-union skeletal fractures in 28 patients. One year follow-up found that all participants did not show any adverse events and 26 of them showed radiographic signs of bone healing [134]. In a pilot trial, autologous alveolar BM-MSCs seeded on serum cross linked scaffold and subjected to osteogenic differentiation caused increased bone density when applied to maxillary cystic defects [135].

BM-MSCs have been used in periodontal regeneration both experimentally and clinically. Following autogenous transplantation of BM-MSCs into class III furcation defects in dogs, they differentiated into cementoblasts, osteoblasts, osteocytes and fibroblasts [136] and enhanced regeneration of PDL, bone and cementum [137]. In a phase I/II clinical trial of autologous BM-MSCs, platelet rich plasma (PRP) and 3D woven fabric composite scaffold showed improved CAL, PD and bone growth over 36 months of follow up in 10 periodontitis patients not showing signs of systemic disease. Cells were isolated through iliac bone marrow aspiration [138], a high risk procedure that its acceptance and justification in periodontitis patients could be quite controversial.

#### 6.1.3. PDL-MSCs vs. BM-MSCs for Periodontal Regeneration

Studies comparing PDL-MSCs and BM-MSCs in the context of periodontal regeneration are fairly few and show variable results. PDL-MSCs formed more colonies but took longer time to differentiate into osteocytes and chondrocytes when compared to BM-MSCs [139]. A similar conclusion regarding colony forming efficiency was reached using patient matched jaw bone BM-MSCs and PDL-MSCs. Both cell types had similar proliferation rates initially but PDL-MSCs outpaced BM-MSCs later [140]. A pioneer study used composite cell sheets that combined both cell types and showed higher expression of bone markers compared to cell sheets with a single cell type upon transplantation in mice [141]. Comparing cell sheets of autologous BM-MSCs and PDL-MSCs transplanted to canine periodontal defects, more cementum, well oriented PDL fibres and alveolar bone were observed with PDL-MSCs sheets [142]. Autologous BM-MSCs formed more bone on both short and long terms compared to PDL-MSCs in a canine peri-implant defect model [143].

BM-MSCs potential for periodontal regeneration in an inflammatory microenvironment using TNF-α was compared to that of PDL-MSCs. One study showed that TNF-α reduced proliferative potential of BM-MSCs and mineralization of PDL-MSCs [144]. A second study concluded that both cell types had similar proliferation rates and colony formation efficiencies, however PDL-MSCs displayed weaker osteogenic potential that was further inhibited by TNF-α [145].

#### 6.1.4. PDL-MSCs Isolated from Diabetic Patients and/or Cultured under Diabetic Conditions

There is limited literature on characterization of PDL-MSCs isolated from diabetic patients. PDL-MSCs and G-MSCs were harvested from impacted third molars extracted from healthy and controlled T2DM patients (HbA1c below 7% at time of study entry) and compared in terms expression of CD45, CD90 and CD105 as well as cell proliferation. Both PDL-MSCs and G-MSCs from healthy donors showed higher proliferation rates compared to diabetic counterparts. Moreover, PDL-MSCs had more proliferative potentials compared to G-MSCs in both healthy and diabetic conditions. All isolated cell populations showed positive expression of CD90 and CD105 and negative expression of CD45 without details of significant difference between diabetic and non diabetic cells [146]. Another study compared PDL-MSCs from teeth with periodontitis from diabetic patients to those extracted from non-diabetics both sound and with periodontitis. The diabetic cohort was well controlled with HbA1c levels range 6.5–7.5%. The authors concluded that periodontitis and diabetes PDL-MSCs group had the lowest osteogenic and adipogenic potentials followed by periodontitis only group and the best potentials were observed in PDL-MSCs isolated from sound teeth of non-diabetics [147]. The third study isolated PDL cells from extracted teeth of long standing insulin dependent diabetic mellitus (IDDM) patients. These patients had age range 38–45 yrs with diabetes duration ranging from 5 to 10 yrs and thus their diagnoses as T1DM or T2DM would need further confirmation. Diabetic teeth donors were well controlled at time of extraction although history of episodes of poor control was reported. Extracted teeth and subsequently isolated cells in both diabetic and healthy groups were a mix of sound and periodontally involved third molars, but the authors confirmed that this had no statistical influence on the results. This investigation concluded that diabetic PDL cells grew at similar rates compared to cells from healthy donors. However, they had lower ALP activity and lesser rate of mineralised nodules formation than healthy cells [148]. Thus it seems that overall PDL-MSCs from diabetics have a trend of lower proliferation rate and osteogenic potentials, but because of the small number of studies it would be hard to draw firm conclusions. Additional research is needed to fully understand the impact of DM on PDL-MSCs numbers and differentiation potentials.

Several studies have cultured PDL-MSCs under diabetic conditions to simulate pathological changes attributed to diabetes in vitro and their results are summarized in Table 1. Their general conclusion was that osteogenic potentials are reduced. Most of these studies used high glucose culture conditions [149,150] and some used AGEs as the later would reflect the chronicity of changes in diabetic microenvironment rather than mere HG [151,152]. In fact, diabetic microenvironment could be more complicated than short term exposure to HG [148]. For instance, macrophages isolated from rats with short term diabetes completely recovered to normal levels of cytokines production after 5 days of normoglycemic culture, while cells from rats with long term diabetes showed only partial recovery. Because both groups had higher level of serum glucose, this differential recovery could be attributed to the fact that only long term diabetic rats had higher serum lipids. This conclusion is further supported by the finding that non diabetic hyperlipidemic animals also displayed the incomplete recovery [153]. Another rather interesting finding is that the HG concentrations used in these studies are far higher than the ones found physiologically, even in severely uncontrolled diabetics. Where experimental glucose concentration of 5.5 mM represent normoglycaemia or plasma glucose level of 100 mg/dL, 24 mM glucose should denote hyperglycaemia or plasma glucose level of 432 mg/dL (uncontrolled diabetics have BGL of 200 mg/dL or above) [149]. With many studies using glucose concentrations as high as 30 mM (around 550 mg/dL) to simulate hyperglycaemia [150,154], this poses questions about the physiologic relevance of the used glucose concentrations.

The results of studies of PDL-MSCs from diabetics reviewed above come in agreement with studies on PDL-MSCs isolated from healthy subjects and cultured under induced diabetic conditions. This suggests that this experimental approach could mirror pathological changes seen in diabetic PDL-MSCs. However, expression of IGF axis genes in PDL-MSCs was not investigated in either model. It is worth mention that in some of these studies authors referred to the isolated cells as PDL cells or fibroblasts where their ‘stemness’ was not completely verified using multilineage differentiation or surface markers expression. Although several 3D and multi-layered models were developed to simulate and investigate PDL regeneration [155,156,157,158,159], none of them examined this under diabetic conditions.

#### 6.1.5. BM-MSCs Isolated from Diabetic Patients and/or Cultured under Diabetic Conditions

A review on the characterization of MSCs from diabetics revealed most of this work focused on adipose tissue MSCs and revealed little consensus regarding the proliferation efficiency, viability, immunophenotyping, multipotency and homing of diabetic MSCs. The authors concluded that molecular basis and signaling molecules regulating diabetic MSCs still need to be understood [176]. The isolation and characterization of BM-MSCs from T2DM patients in majority of published studies was performed in the context of evaluating their potentials to differentiate into insulin producing cells to reverse diabetes [177,178]. One of the earliest of these studies was conducted in 2009 on BM-MSCs isolated from T2DM patients undergoing cardiac bypass surgery with age range of 15 to 80 years. The study concluded that diabetic BM-MSCs had multilineage differentiation potentials and expressed MSCs markers with cells from uncontrolled, long standing or elderly diabetics showing weaker proliferation. There was, however, no ‘healthy’ BM-MSCs included in the study as controls [177]. In another study, diabetic donors were insulin dependent T2DM patients with mean HbA1c of 11% and age range of 43–55 years. The samples were bone marrow aspirates (BMA) from the iliac crest. No data on other comorbidities were available and both cell populations had similar proliferation rates [178]. In a study comparing healthy, ischemic and ischemic diabetic BM-MSCs cells from patients with critical limb ischemia (CLI), no difference in clonogenic, osteogenic, adipogenic or angiogenic potentials was reported. CLI cells showed lower proliferation rates irrespective of patients diabetic status when compared to healthy controls [179].

Recently, more studies have extensively compared diabetic and non-diabetic BM-MSCs. One of these concluded that multilineage differentiation (including osteogenic), immunomodulatory properties, transcriptomic data, surface markers expression and number of BM-MSCs in the starting material were comparable in both cell populations. These parameters however were reduced in both cell types by passaging cells in vitro [180]. Another investigation reached similar conclusions in addition to diabetic BM samples having fewer MSCs and osteoblasts as measured by colony forming unit–fibroblast (CFU-F) and colony forming unit-osteoblasts (CFU-O) assays [181]. The combined results of both studies resonated well with a clinical study that used autologous bone marrow concentrate (BMC) in treatment of tibial non unions in 54 diabetic patients (*n* = 42 for T2DM and *n* = 12 for T1DM) and equal number of non-diabetic matched control patients. BMA of both cohorts had similar number of MSCs as evident by CFU-F, but this did not equate to having similar treatment outcomes as diabetic patients needed more time for healing and had smaller callus. Treatment failure in diabetics was rationalized to having more comorbidities and lower number of MSCs compared to diabetics with successful treatment outcomes. Consequently, the study recommended the use of larger volumes of BMA and higher number of transplanted cells in diabetics [182].

So far from the aforementioned studies, it would be fair to conclude that T2DM influence on BM-MSCs may not be that drastic. Nonetheless, the results of studies investigating BM-MSCs under in vitro diabetic culture conditions using HG, AGEs, and serum of diabetic patients show different findings (Table 2). In general, HG induced lower proliferation rates while HG combined with LPS and serum of T2DM patients had the opposite effect. AGEs were used only in 2 studies, with one reporting increased expression of OPG, RANKL and RAGEs and the other concluding reduced proliferation of investigated cells. But, as mentioned in PDL-MSCs section, these methodologies have their inherent limitations since many more factors are in play to determine how T2DM affects MSCs in general and this may explain the variable results obtained compared to studies on diabetic BM-MSCs.

### 6.2. Growth Factors in Regenerative Periodontal Therapy

Growth factors (GFs) are natural proteins controlling fundamental cell activities such mitotic division (proliferation), migration, metabolism and differentiation consequently influencing tissue repair and regeneration following injury [193]. A number of GFs are expressed in periodontal tissues including insulin-like growth factors (IGF), platelet derived growth factor (PDGF) and bone morphogenic proteins (BMPs) [194]. The following GFs have been approved by United States Food and Drug Administration (FDA) for periodontal regeneration: Amelogenins (including EMD), PDGF, BMPs, FGF, Teriparatide hormone and platelets concentrate [195]. However there are no robust data on the use of these factors in diabetic patients and some studies reported failure of osteogenic differentiation of MSCs stimulated with BMPs [196]. In addition, lower serum levels of IGF axis proteins were reported even in well controlled T2DM patients and were associated with higher risk of CVD [197] and were also reported in patients with both impaired glucose tolerance and T2DM as a marker of reduced insulin sensitivity [198]. Systemic administration of members of IGF axis was reported to improve glycemic control in T2DM patients [199] as well as insulin sensitivity and pancreatic β-cell functions [200] making IGF proteins promising candidates as therapeutic modalities for obesity, insulin resistance and diabetes [201,202]. Taken altogether, utilizing IGF axis proteins in periodontal regeneration in diabetics could have the extra benefit of improving insulin sensitivity in periodontal tissue compared to other GFs. Although BMPs for instance were proposed as potential insulin sensitizers [203], it would be fair to assume IGFs would exert such effects more efficiently due to their structural and functional similarity to insulin [204,205].

#### 6.2.1. IGF Axis

The insulin-like growth factor (IGF) axis consists of two ligands (IGF-1 and IGF-2), their corresponding receptors (IGF-1R and IGF-2R) as well as six circulating binding proteins (IGFBP-1 to IGFBP-6) (Figure 1). This axis plays a major role in development and maintenance of mineralized tissues [9]. IGF axis is regulated by growth hormone (GH) which is secreted from the anterior pituitary gland-under control of growth hormone-releasing hormone (GHRH) and somatostatin-and binds to its widely distributed receptors. Upon GH binding to hepatocytes, they secrete around 75% of circulatory IGF-1 while the locally produced IGF-1 by other tissues constitute around 25% of serum levels [6]. The IGFBPs are peptides of approximately 260 amino acids that bind IGFs with both IGF dependent functions (transportation in serum, control of vascular efflux and clearance, prolonging half-life, serving as reservoirs, tissue specific directing and regulation of receptor interaction) and independent functions as well [206]. The later include regulating gene transcription, angiogenesis, autophagy and cell senescence [207]. Expression of IGF and IGFBPs in calcified tissues is under regulation of factors classically known for prompting bone formation such as vitamin D3, GH, parathyroid hormone and IGF-1. Although the total effect of IGF axis is promoting bone development, the expression profile and functions of its proteins on the molecular level are complex depending on stage of cells differentiation and matrix mineralization [208] which is outlined in the next sections.

The IGF axis consists of IGF1 (red circle), IGF2 (black circle), IGF1R (red receptor), IGF2R (black receptor), IGF2R (black receptor) and 6 binding proteins IGFBP1 to 6. IGF1 and IGF2 bind IGF1R with IGF1 having higher affinity. Both ligands can bind hybrid receptor IGF1R/IR (green receptor). However, only IGF2 can bind IR and IGF2R and insulin (yellow circle) can bind IR and hybrid receptor IGF1R/IR. IGFBP3 and 5 bind to acid labile subunit (ALS, yellow hexagon) to form tertiary complexes. This figure is designed using Servier Medical art (http://smart.servier.com, accessed on 28 September 2020). Servier Medical Art by Servier is licensed under a Creative Commons Attribution 3.0 Unported License.

#### 6.2.2. Roles of Different IGF Axis Proteins in Osteogenesis

IGF-1 is the most abundant growth factor in bone microenvironment through local production by bone cells or remotely by hepatocytes and then transportation to bone tissues [209]. IGF-1 serum levels are also indicative of bone mineral density [210]. IGF-1 increases RANKL production by osteoblasts leading to osteoclasts activation and bone resorption. However, this is part of the overall role of IGF-1 in bone remodeling where bone resorption precedes deposition, with a net anabolic effect on bone [211].

IGF-1R is structurally similar to insulin receptor (IR) and both can bind insulin, IGF-1 and IGF-2 with variable affinity and the mechanism of IGFs binding to IGF1-R is assumed comparable to insulin binding IR [212]. The IGF-2R/mannose 6 phosphate (M6P) receptor is a type 1 transmembrane glycoprotein with the main action of suppressing IGF1-R signaling through binding to excess extracellular IGF-2. IGF2/IGF2-R interaction is key for normal development and is involved in carcinogenesis as well, with IGF2-R reported as an oncogene in some cancers and as tumor suppressor gene in others [213].

IGFBP-1 is a secretory protein with serum levels lower in patients with glucose intolerance and positively correlated with insulin sensitivity [214,215]. IGFBP-1 phosphorylated form has more affinity to IGF1 than the non-phosphorylated one and it was shown to have both stimulating and inhibitory effects on IGF-1 [216]. IGFBP-2 effects on IGFs action are mainly inhibitory and serum levels of IGFBP-2 increased with aging in both men and women and were associated with lower bone mineral density (BMD) [217] and with expression of bone resorption markers [218]. IGFBP-2 in gingival crevicular fluid was higher in periodontitis patients and correlated positively with CAL and BOP [219]. However, on the cellular level, IGFBP-2 along with IGF-1 constitute key factors for osteoblasts differentiation [220]. IGFBP-2 was upregulated during osteogenic differentiation of DP-MSCs and enhanced IGF-1 induced matrix mineralization of these cells as well [7]. IGFBP-3 is the main IGFBP in serum with 75–80% of IGFs in serum form ternary complexes with IGFBP-3 (and less commonly IGFBP-5) and acid labile subunit (ALS) [221]. Serum levels of IGFBP-3 were linked with higher BMD in healthy men [222], but also with vertebral fractures in postmenopausal women [223] and higher CAL and more teeth loss in patients with periodontitis [224].

IGFBP-4 is 237 residue protein with mostly inhibitory effects on IGF-1 and IGF-2 and its overexpression in bone tissue lead to impaired bone formation and growth. However, some studies have shown anabolic effect of IGFBP-4 where its systemic administration increased expression of ALP and OCN in mice bone and serum [225]. IGFBP-5 is found in multiple tissues and is the most ample IGFBP in bone. IGFBP-5 can bind ECM which offers protection against IGF-1 degradation, represent a reservoir for IGF-1 and potentiates its effects. IGFBP-5 stimulatory and inhibitory action on IGF have been reported [226]. IGFBP-6 expression is regulated by vitamin D, retinoic acid, IGFs, glucocorticoids, Wnt and Hedgehog pathways [227]. IGFBP-6 functions include inhibition of IGF-2 dependent cell division, migration, differentiation and survival in addition to inhibiting cell proliferation and stimulating apoptosis independently of IGF-2 [228]. IGFBP-6 also holds intracellular function through binding to nuclear and possibly mitochondrial receptors as well [229].

#### 6.2.3. Expression of IGF Axis in the Periodontium

With exception of IGF-2R, all IGF axis members are expressed in cementum and PDL tissue in variable degrees. Immunohistochemical analysis showed that cementum Sharpey’s fibers show strong expression of IGF-1, IGF-2 and IGFBP-5 while PDL tissue ECM (not cells) displayed high immunoreactivity of IGF-1 and IGFBP-6. Interestingly, PDL cells showed immunoreactivity only to IGF-1R and cementum cells stained positive only with IGFBP-2 [230]. This is consistent to some extent with the work of Reckenbeil et al. [231] that investigated IGF axis expression in PDL cells excluding IGF2 and IGF2-R. PDL cells cultured under basal conditions barely expressed mRNA of IGF-1 and IGFBP-1. However, IGF-1R was clearly expressed at both gene and protein levels.

Nevertheless, different expression profiles were concluded in other studies. Compared to gingival fibroblasts, PDL cells overexpressed IGF-1R and IGFBP-5 on gene level [232]. PDL cells were also found to express mRNA of IGF-2 and IGFBP-6 and both show time dependent increase at gene level and decrease at protein level, but this study did not examine other IGF axis genes and thus no firm conclusions can be drawn about their relative expression patterns [233]. Whether these expression patterns would be different in PDL-MSCs from diabetics is still to be explored.

#### 6.2.4. IGF Axis in Periodontal Regeneration

Members of IGF axis are being used for experimental treatment of periodontal diseases/bone defects in animal models since late 1980s. For instance, IGF-1 combined with PDGF was applied to roots surfaces of teeth with periodontitis in beagle dogs following OFD. This procedure prompted formation of new cementum and bone with the later lined with a continuous layer of osteoblasts compared to control sites which healed with long junctional epithelium (LJE) [234] and similar results were induced in monkeys [235]. However, when PDGF and IGF-1 were tested individually, PDGF solely could provoke the periodontal regeneration unlike IGF-1. Adding IGF-1 to PDGF significantly stimulated the positive effect of PDGF [236]. Both GFs were used on dentine chips from roots of human extracted teeth with periodontitis. Dentine chips were treated with tetracycline, GFs and the cultured PDL fibroblasts. Consistent with the above studies, IGF-1 alone did not evoke a significant difference, but PDGF did and this was enhanced by adding IGF-1, with tetracycline treatment causing little effect [237].

One study using collagen sponges loaded with a combination of IGF-2, FGF and TGF-β and applied on alveolar bone defects in dogs recorded higher bone formation in controls that received collagen sponges with vehicle only. The authors propose that the collagen sponge could have interfered with the wound healing process [238]. Another possible explanation is unknown interactions of the 3 GFs applied simultaneously with little spatial or temporal control of release. IGF-1 surgically applied to experimental class II furcation defects in beagle dogs improved bone, cementum and PDL regeneration. However, the results were drastically improved when IGF-1 was incorporated into a local drug delivery system of dextran-co-gelatine micro spheres [114]. Systemic IGF-1 improved glycaemic control and increased rate and height of bone formation in diabetic rats following teeth extraction. Although alveolar bone remodeling after dental extraction is not a periodontal bone defect, these results can still be plausible for periodontal regeneration in diabetics [239]. In swine models of periodontitis, locally applied IGFBP-5 improved PD, CAL and new bone formation [240]. Clinically, a phase I/II clinical trial showed that a combination of rh-IGF1 and rh-PDGF produced significant bone fill in angular bone loss including class II furcation involvement [241]. Rh-IGF-1 used with TCP bone graft and PLGA membranes in clinical surgical treatment of two wall intraosseous defects improved PD, bone loss and CAL and combining rh-IGF-1 with rh-VEGF produced even better results (363). The outcomes of these investigations were in part explained by in vitro studies looking closely at how IGF axis proteins influence PDL cells as shown in the next section.

#### 6.2.5. Effect of IGF Axis on PDL Cells

Controversial findings were reported about PDL cells response to IGF-1 stimulation. PDL fibroblasts and to a lesser extent gingival fibroblasts showed higher proliferation in response to IGF-1 but not GH, possibly due to lack of immediate effect of GH on these cells. This exposure to IGF-1 also influenced expression of proteoglycans, with down regulation of decorin and upregulation of versican and biglycan, indicating potential role of IGF-1 in ECM homeostasis of PDL tissues [242]. A second study reported increased proliferation in response to EMD alone and in combination with IGF-1 but not IGF-1 solely. Both GFs and their combination had no effect on cellular migration, adhesion or collagen production [243]. Rh-IGF1 stimulated DNA synthesis and proliferation of PDL cells, and IGF-1 expression was upregulated on both gene and protein levels using EMD. This was confirmed when anti-h IGF-1 antibodies reversed the EMD-mediated growth of PDL cells and it is thought that EMD effects could be conveyed in part by endogenous production of IGF-1 by these cells [244]. This is different from the results of cDNA array analysis where EMD upregulated a number of GFs in PDL cells including PDGF, BMP-1 and 4 but not IGF-1 [245]. One possible explanation is the difference in concertation and duration of EMD treatment, where the first study used 50 µg/mL EMD for a max of 2 days but the second study used 100 µg/mL EMD for 4 days.

Exogenous IGF-1 and IGF-2 induced proliferation of both preconfluent and confluent PDL cells under basal conditions. However, IGF-1 treatment was associated with higher expression of OCN in both preconfluent and confluent cells and higher ALP activity only in preconfluent PDL cells. These effects were not matched by IGF-2 (IGF-2 induced only ALP activity in confluent cells). This is consistent with IGF-2 acting as a promitogenic agent maintaining a pool of undifferentiated cells while IGF-1 is more committed to PDL cells differentiation [246]. Opposing results were reported for PDL cells cultured under osteogenic media, with adding rh IGF-1 increasing mineral deposition despite decreasing ALP activity and having no significant influence on OCN expression in PDL cells [231]. Another interesting finding is that osteogenic media suppressed IGF-1 and upregulated IGF-2 mRNA expression, which may indicate that their roles change in different contexts or culture conditions [247]. IGF-1 could promote PDL-MSCs proliferation and its optimized concertation increased cellular organelles, Alizarin red staining and ALP activity. Moreover, IGF-1 induced the upregulation of Runx2, OCN and OSX expression at both mRNA and protein levels under osteogenic media [248]. Similar results were obtained by combining IGF-1 with platelet rich fibrin (PRF) [249]. IGF-1 also reduced apoptosis and caspase-3 expression in PDL cells [250].

These effects of IGF-1 on PDL-MSCs were most likely mediated through IGF-1R, as silencing this receptor inhibited the IGF-1 induced proliferation and migration of PDL-MSCs. IGFR-1 silencing also decreased IGF-1 expression by PDL-MSCs [246] and selective blockers of IGF-1R kinase reduced calcium deposits of PDL cells, confirming its role in regulating IGF-1 mediated osteogenesis [247].

IGF-1 was also upregulated in PDL cells under hypoxic conditions, possibly to decrease apoptosis and promote ROS scavenging through PI3K/Akt/mTOR signaling pathway [251]. Similar influence was caused by intermittent compressive stress simulating occlusal load via TGFβ pathway. Interestingly, hypoxia attenuated this upregulation, suggesting that IGF-1 may not be upregulated by occlusal forces in hypoxic deep periodontal pockets which may contribute to progression of periodontitis [252]. Single exposure to TGFβ upregulated IGF-1 and osteogenesis in PDL cells, while the opposite was observed after repeated exposures to TGF-β. In the later situation, IGF-1 induction rescued the deteriorated osteogenic differentiation of PDL cells, suggesting it could be a valuable tool for promoting bone regeneration under chronic inflammatory conditions [253].

The effect of IGF-1 on PDL-MSCs was compared to other dental MSCs. IGF-1 was expressed at higher levels in PDL-MSCs than DP-MSCs. However, when both cell types underwent osteogenic/odontogenic induction, IGF-1 was upregulated in DP-MSCs but down regulated in PDL-MSCs. With the overall changes in gene expression in PDL-MSCs being more complex (higher number of altered genes with mostly down regulation), this possibly reflects the distinctive nature of both cell populations. DP-MSCs primarily differentiate into odontoblast for pulp maintenance, while PDL-MSCs differentiate into 3 different cell population for homeostasis of both soft and hard tissues forming periodontium [254].

Regarding IGFBPs, PDL-MSCs express higher levels of IGFBP-5 compared to BM-MSCs, with osteogenic culture conditions upregulating its expression in both cell types. IGFBP-5 knockdown decreased ALP activity and alizarin red staining in both PDL-MSCs and BM-MSCs as well as inhibiting the expression of OSX, ON and OCN in PDL-MSCs. Moreover, IGFBP-5 silencing led to increased expression of IL-6 and IL-8 in PDL-MSCs [255]. IGFBP-5 silencing similarly reduced proliferation, migration, chemotaxis and osteogenic potentials of PDL-MSCs treated with TNF-α, which were all rescued by adding rh-IGFBP5 [240]. Considering the fact that IGFBP-5 seems to be the only IGFBP tested in animal models of periodontitis, it would be a promising candidate for future clinical trials in addition to IGF-1 and IGF-2.

#### 6.2.6. Effect of IGF Axis on BM-MSCs

IGF-1 stimulated proliferation and osteogenic differentiation of BM-MSCs with upregulation of ALP, Runx2, OCN and calcium handling proteins [256]. IGF-1 stimulation of BM-MSCs in osteogenic cultures lead to significant increase in ALP activity while BMP-7 lead to non-significant decrease in ALP activity [257]. Another study found that IGF-1 induced both ALP activity and minerals formation in BM-MSCs superior to BMP-7, even if the difference was not statistically significant suggesting IGF-1 represents a more promising candidate for clinical application in bone fractures [258]. IGF-1 mediated osteogenesis was linked to activation of MAPK and PKD signalling pathways and subsequent upregulation of Osx [259]. IGF-1 stimulatory action extended to aging BM-MSCs as well [260].

Although IGF-2 has no direct effect on pregnancy associated plasma protein A (PAPP-A) or its inhibitor expression in primary human osteoblasts, IGF-2 is necessary for proteolysis of IGFBP-4 functionally. The IGF-2 dependent proteolysis of IGFBP-4 and subsequent higher IGF-1 bioavailability were increased by pre-treating human osteoblasts with TGF-β which stimulated PAPP-A and suppressed PAPP-A inhibitor. This means that IGF-1 bioavailability in human BM-MSCs is regulated by both IGF-2 and TGF-β [261]. IGFBP-4, known for its inhibitory effects, was upregulated in passage (P) 10 BM-MSCs which showed higher apoptosis and senescence compared to P1 cells. When rIGFBP-4 was added to P1 cultures, they showed similar changes to the ones seen in P10 cells. IGFBP-4 blockers reversed the changes seen in P10 BM-MSCs, suggesting they can be a valuable tool for treating BM-MSCs senescence prior to autologous transplant to periodontal defects [262]. The role of IGF axis in stem cell- based regeneration under diabetic conditions is still an under investigated area and represents a research gap at the moment. However, very few studies have shown that IGF-1 helped improve wound healing in STZ diabetic rats [263] and augmented BM-MSCs regenerative potentials in STZ diabetic mice [264].

## 7. Conclusions

Regenerative therapy using MSCs and/or growth factors represents the future for bone regeneration including dental and periodontal applications. The current non surgical and surgical periodontal treatments have their strengths but also their limitations, most notably in cases of suprabony pockets. Another shortcoming is lack of well-designed clinical trials of regenerative surgeries in patients with T2DM who represent a growing proportion of dental patients. Further research is needed to understand the influence of diabetes on native PDL-MSCs on the molecular level and whether using growth factors such as IGFs or their binding proteins would enhance periodontal regeneration in diabetics, in particular IGF-1, IGF-2 and IGFBP-5. Additionally, more work is required to optimize the delivery form of growth factors into periodontal defects via non-surgical approach because of poor wound healing in diabetics. Pretreatment of diabetic BM-MSCs before autologous transplant to periodontal bony defects using IGFs or IGFBPs to enhance their proliferative and osteogenic potentials can be another treatment option as well. Still, using signaling molecules to boost endogenous PDL-MSCs would be more practical and cost effective. Preclinical models followed by well-designed clinical trials of both approaches are necessary for full transition to dental clinical settings.

## Figures and Tables

**Figure 1 bioengineering-08-00202-f001:**
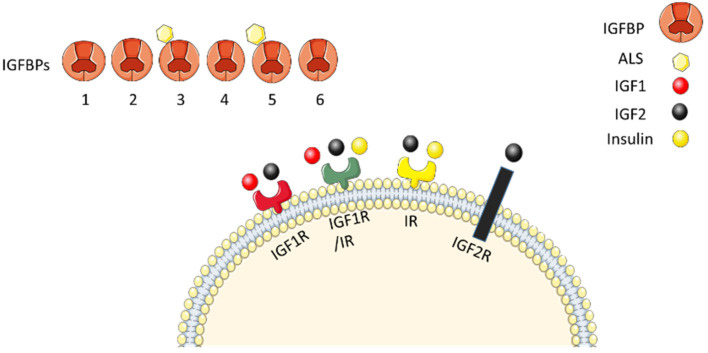
Components of IGF axis.

**Table 1 bioengineering-08-00202-t001:** Effect of diabetic culture conditions on human PDL-MSCs.

#	Study	Diabetic Cond	Ca Nodules (AR Staining)	ALP Expression	ALP Activity	Osteogenic Transcription Factors		Osteogenic Markers		NF-ĸβ	Others
						RUNX2	OSX	OCN	OPN	Expression	
1	Zhen et al. [160]	HG	↓	NR	NR	↓	↓	↓	NR	NR	↑miR-31
2	Liu et al. [161]	HG	NR	NR	↓	NR	↓	↓	↓	NR	↑DNA methylation
3	Kato et al. [149]	HG	↓	NR	↓	↑	NR	↓	NR	↑	↓ Proliferation
											↓ Viability
											↑ IL-6 and IL-8 expression
											Morphology NC
4	Guo et al. [150]	HG	↓	↓	NR	↓	↓	NR	NR	NR	↓ Proliferation
											↓ Cells in S and G2/M phases
5	Zheng et al. [154]	HG	NR	NR	↓	↓	↓	NR	NR	NR	↓ Proliferation
6	Zhan et al. [162] *	HG	NR	NR	NR	NR	NR	NR	NR	NR	↑ RAGEs expression
											↓Proliferation
7	Kim et al. [163]	HG	NR	↓	↓	↓	↓	NR	↓ **	NR	↓COL1A1 **
8	Yan et al. [164]	HG	↓	NR	↓	NR	NR	NR	NR	NR	↓ Migration
											↑ ROS
9	Deng et al. [165]	HG	↓	↓	NR	↓	NR	NR	↓	NR	↑ Adipogenic differentiation
10	Bhattarai et al. [166] *	HG	↓	NR	NR	NR	NR	NR	NR	NR	↓ Proliferation
											↑ ROS
11	Kim et al. [167] *	HG	↓	NR	NR	NR	NR	NR	NR	NR	↓ Proliferation
											↓ Viability
12	Liu et al. [168] *	HG	NR	NR	NR	NR	NR	NR	NR	NR	↑ Apoptotic cells
											↑ Caspase 3 activity
13	Luo et al. [169]	HG	NR	NR	NR	NR	NR	NR	NR	NR	↓ Proliferation
											↓ DNMT activity
											↑ TNFR-1 expression
14	Zhang et al. [170]	HG	NR	NR	NR	NR	NR	NR	NR	NR	↑ RANKL
											↓ OPG
15	Seubbuck et al. [171] *	HG	↑	NR	↑	NR	NR	NR	NR	NR	↑ Proliferation
											↑ Expression of Nanog, Oct4, Sox2, CD166, Periostin and β-Catenin
16	Xu et al. [29] *	AGEs	NR	NR	NR	NR	NR	NR	NR	↑	↓ Viability
											↑ IL-6 and IL-8 expression
											↑ ERS
17	Guo et al. [151]	AGEs	↓	NR	NR	NR	NR	NR	NR	NR	↑ RAGEs
											↑ ROS
18	Mei et al. [172] *	AGEs	NR	NR	NR	NR	NR	NR	NR	NR	↓ Viability
											↑ Apoptosis
											↑ Autophagy
											↑ ROS
19	Wang et al. [173]	AGEs	↓	NR	↓	↓	NR	NR	↓	NR	↓ Proliferation
20	Zhang et al. [152]	AGEs	↓	↓	↓	↓	↓	↓	↓	NR	↓ COL1 expression
											↓ BSP expression
21	Fang et al. [174]	AGEs	↓	↓	↓	↓	NR	↓	NR	NR	↓ Proliferation
											↑ ROS
											↑ Apoptosis
											↑ Mitochondrial damage
22	Zheng et al. [10] *	HG + TNF-α	NR	NR	NR	NR	NR	NR	NR	↑	↑ RANKL expression
23	Zhu et al. [175]	HG + TNF-α	NR	NR	NR	NR	NR	NR	NR	NR	↑ TNFR-1 expression
											↓ Cell viability

(↓): reduced compared to control culture media. (↑): increased compared to control culture media. * PDL cells/fibroblasts. ** Protein level only. AGEs: advanced glycation endproducts. ALP: alkaline phosphatase. AR: Alizarin red. BSP: bone sialoprotein. COL1A1: collagen 1 A1. DNMT: DNA methyltransferase. ERS: endoplasmic reticulum stress. HG: high glucose. IL-1β: interleukin 1β. IL-6: interleukin 6. IL-8: interleukin 8. NC: no change. NR: not reported. OSX: Osterix. OCN: Osteonectin. OPN: Osteopontin. RAGEs: receptors of advanced glycated endproducts. RANKL: receptor activator NF-ĸβ ligand. ROS: reactive oxygen species. Runx2: Runt related transcription factor 2. TNF-α: tumour necrosis factor α. TNFR-1: tumour necrosis factor-alpha receptor-1.

**Table 2 bioengineering-08-00202-t002:** Effect of diabetic culture conditions on human BM-MSCs.

#	Study	Diabetic Cond	Ca Nodules (AR Staining)	ALP Expression	ALP Activity	Osteogenic Transcription Factors		Osteogenic Markers		Others
						RUNX2	OSX	OCN	OPN	
1	Ying et al. [183]	HG	↓	↓	↓	↓	↓	↓	NR	↓COL1 and BMP2 expression
										↓PI3k and Akt expression
										↑ ROS
2	Chang et al. [184]	HG	NR	NR	NR	NR	NR	NR	NR	↓ PD time
										↑ Senescence
										↑ Autophagy
3	Li et al. [185]	HG	↑	NR	NR	NR	NR	NR	NR	↓Proliferation (25 mM, long term)
										↓ Apoptosis (40 mM, short term)
4	Dhanasekaran et al. [186]	HG	NC	NR	NR	NR	NR	NR	NR	↓ Proliferation
	(late vs early P)									Unchanged morphology, karyotyping and surface markers
5	Shiomi et al. [187]	HG + LPS	↓ (24 mM, 3 wks.)	NR	↓ (12 and 24 mM, 2 wks.)	↑ (8 and 12 mM, 3 wks)	NR	↓ (12 and 24 mM, 3 wks.)	NR	↑ Proliferation
	(purchased BM-MSCs)					↓ (24 mM, 3 wks.)				↓ IL-1β, IL-6, IL-8
										(8 and 12 mM, 1 and 2 wks.)
										↑ IL-1β, IL-6, IL-8
										(24 mM, 1 and 2 wks.)
6	Qu et al. [188]	HG + free fatty acid	↓	NR	↓	NR	NR	NR	NR	↑miR-449
7	Bian et al. [189]	HG + palmitic acid	↓	NR	↓	↓	NR	↓	NR	↓ Proliferation
	(BM-MSCs cell line)									↓p38 expression
										↑ ROS
8	Miranda et al. [30] (primary osteoblast like cells from T2DM patients)	HG + AGEs	NR	NR	NR	↓	↓	NR	NR	↑ OPG and RANKL expression
										↓ OPG/RANKL ratio
										↑ RAGEs expression
9	Lu et al. [190]	AGEs	NR	NR	NR	NR	NR	NR	NR	↓ Proliferation
10	Deng et al. [191]	T2DM serum	NR	NR	NR	↓	NR	↓	↓	↑ Proliferation
11	Rezabakhsh et al. [192]	T2DM serum	NR	NR	NR	NR	NR	NR	NR	↑ Apoptosis
										↑ Autophagy
										↓ Chemotaxis
										↓ Angiogenesis

(↓): reduced compared to control culture media. (↑): increased compared to control culture media. AGEs: advanced glycation endproducts. ALP: alkaline phosphatase. AR: Alizarin red. BMP2: bone morphogenic protein 2. BSP: bone sialoprotein. COL1: collagen 1. HG: high glucose. IL-1β: interleukin 1β. IL-6: interleukin 6. IL-8: interleukin 8. LPS: lipopolysaccharides. NC: no change. NR: not reported. OCN: osteonectin. OPG: osteoprotegerin. OPN: osteopontin. OSX: osterix. PD: population doubling. RAGEs: receptors of advanced glycated endproducts. RANKL: receptor activator NF-ĸβ ligand. ROS: reactive oxygen species. Runx2: Runt related transcription factor 2. T2DM: type 2 diabetes mellitus.

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
