# Peer review of "The Effect of Diabetes Mellitus on IGF Axis and Stem Cell Mediated Regeneration of the Periodontium"

_bioengineering, 2021, doi:10.3390/bioengineering8120202_

Round 1

Reviewer 1 Report

The authors have responded to my comments reasonably. 

Reviewer 2 Report

The manuscript could be accepted in the present form, the authors made the required corrections.

This manuscript is a resubmission of an earlier submission. The following is a list of the peer review reports and author responses from that submission.

Round 1

Reviewer 1 Report

The review describes work done on effect of diabetes on regeneration of periodontium. Even though they describe the studies on regenerative therapies in detail, they don't pull the focus back on the effect of diabetes. There are many parts that need to be re-ordered for better flow of thought

Major comments

  1. What does " impaired quantitative and qualitative capabilities of stem cells" mean?
  2. Elaborate on what red, yellow, orange and green complexes mean
  3. The authors have presented a very brief overview of inflammation in the context of diabetes in different tissues/cells. Why is inflammation in different tissues important here? Was this section meant to be an overview of all the inflammatory pathways in Diabetes? If yes, then a lot of pertinent mechanisms have been left out
  4. Elaborate on the effect of inflammatory phenotype of EPCs  
  5. Describe the effect of surgical periodontal therapy on on diabetes pathology? Does the oral microbiome therapy or the surgical therapy alleviate diabetes associated pathology?
  6. The term uncontrolled diabetic animals is used in multiple places. What does it mean?
  7. More information on animal models is necessary - how was diabetes induced? what was the genetic background? 
  8. Authors draw inferences from inflammatory reactions of bone grafts in different parts of the body to state that the response of diabetic patients to periodontal surgeries is unpredictable. These are varied niches and this statement is better supported if the authors can provide evidence of inflammation in the oral cavity from surgeries and its effect on diabetes 
  9. Include a description of how pro-inflammatory and anti-inflammatory phenotypes affect differentiation potentialls of the stem cells - focusing on the mechanisms
  10. Have 3D differentiation models been developed? Does it improve differentiation of cells derived from diabetes patients?
  11. Explain how the osteogenic and adipogenic potentials are calculated
  12. Authors have to justify why they focus on IGF axis even though expression of IGF in PDL-MSC from diabetic patients is not known - does IGF/ IGFR expression change in periodontal tissue in diabetes and how does that affect diabetes pathogenesis (inflammation etc)?
  13. Elaborate on differences in IGF influence on BM-MSCs derived from diabetic patients 

Author Response

Please see the attachemnet 

Reviewer 2 Report

The article addresses a niche in an interesting and extensively studied subject (the periodontium-diabetes relationship).

 But the editing is careless, the text abounds in typos. For a better academic value, grammar and written check is required!!!

E.g.:

..the hroizintal pattern of bone loss [55]. In fact...

....alone [59]. This means that while the benefits of regenertive periododntal....

.....minmally inavsive periodntal surgery with or without EMD was tested in T2DM and non....
....diaebtic elderly pateints with infrabony pockets. There was no signficant differnce....

...realtively large numbers. This means minimum need for expansion and verstaillity of applications.....
....icnlduing banking for future use [83].

And others like that.

Author Response

Response: We thank the reviewer for their comments, and we have carried out a thorough proof reading of the manuscript and corrected all typographical and grammatical errors we have found. 

Reviewer 3 Report

This manuscript described the association between periodontal disease and diabetes from the point of view of periodontal regeneration using mesenchymal stem cells and insulin-like growth factor. The theoretical concept is adequate and 
cited references support it.

Author Response

Response: We thank the reviewer for their encouraging comments.